# Estimation of Peanut Southern Blight Severity in Hyperspectral Data Using the Synthetic Minority Oversampling Technique and Fractional-Order Differentiation

Heguang Sun [1,2], Lin Zhou [3], Meiyan Shu [1], Jie Zhang [2], Ziheng Feng [2], Haikuan Feng [2], Xiaoyu Song [2], Jibo Yue [1,*] and Wei Guo [1,*]

1 College of Information and Management Science, Henan Agricultural University, Zhengzhou 450002, China; sunheguang@henau.edu.cn (H.S.); 13383712711@163.com (M.S.)
2 Information Technology Research Center, Beijing Academy of Agriculture and Forestry Sciences, Beijing 100094, China; zhangj0216@163.com (J.Z.); fengziheng@stu.henau.edu.cn (Z.F.); 2020201097@stu.njau.edu.cn (H.F.); songxy@nercita.org.cn (X.S.)
3 College of Plant Protection, Henan Agricultural University, Zhengzhou 450002, China; zhoulinhenau@163.com
* Correspondence: yuejibo@henau.edu.cn (J.Y.); guowei@henau.edu.cn (W.G.)

**Abstract:** Southern blight significantly impacts peanut yield, and its severity is exacerbated by high-temperature and high-humidity conditions. The mycelium attached to the plant's interior quickly proliferates, contributing to the challenges of early detection and data acquisition. In recent years, the integration of machine learning and remote sensing data has become a common approach for disease monitoring. However, the poor quality and imbalance of data samples can significantly impact the performance of machine learning algorithms. This study employed the Synthetic Minority Oversampling Technique (SMOTE) algorithm to generate samples with varying severity levels. Additionally, it utilized Fractional-Order Differentiation (FOD) to enhance spectral information. The validation and testing of the 1D-CNN, SVM, and KNN models were conducted using experimental data from two different locations. In conclusion, our results indicate that the SMOTE-FOD-1D-CNN model enhances the ability to monitor the severity of peanut white mold disease (validation OA = 88.81%, Kappa = 0.85; testing OA = 82.76%, Kappa = 0.75).

**Keywords:** peanut southern blight; SMOTE; hyperspectral reflectance; machine learning; FOD

## 1. Introduction

Peanut southern blight caused by *Agroathelia rolfsii* has caused huge economic losses in peanut production worldwide [1]. Southern blight, a serious soil-borne fungal disease, develops rapidly under hot and humid conditions. In the early stages of infection, the plant leaves are yellowed, and rough white mycelium clings to the lower stems. As the disease intensifies, the mycelium further radiates to the soil surface, the signs of leaf yellowing intensify, and the infected stems and leaves turn a dark brown color, and ultimately the entire plant dies [2]. There are no peanut varieties with high resistance to southern blight, and the control of southern blight is still based on chemical agents [3]. Therefore, timely and effective monitoring of southern blight incidence is important for peanut control.

Unlike stem rot [4] and leaf spot [5], southern blight is very difficult to monitor at an early stage. Affected plants show slight discoloration in the canopy leaves, and this symptom is not easily detectable [2,6]. Further, as the hot and humid weather intensified, the mycelium inside the infected plants started to spread extensively, and this phase was left very short for our investigations [7,8]. Traditional field surveys are time-consuming, labor-intensive, and not easy to monitor over large areas [9]. In recent decades, the emergence of hyperspectral remote sensing technology has provided new technological means for monitoring diseases in a timely and non-destructive manner [10]. Hyperspectral data is

composed of a large amount of continuous narrow-band data, which, unlike wide-band multispectral data, provides more detailed information about the target. However, these advantages often come at the cost of high dimensionality and large data volumes, which gives birth to the new problem of the curse of dimensionality [11]. As the noise and sparsity of the feature space increase, the performance of the model gradually decreases [12], Small samples and data imbalances can exacerbate this problem [13]. Spectral feature multiclassification models require large amounts of sample data [14]. For our study, the early onset of southern blight was rapid, the monitoring window was short, and some imbalance in the data was inevitable.

Unbalanced data and small sample sizes can seriously affect the robustness and reliability of the model, and even more so for high-dimensional data such as hyperspectral [15]. Over the past decade or so, a variety of approaches have been developed in various fields to cope with this problem [16,17]. This includes various resampling methods, namely oversampling, undersampling [18], as well as some feature selection and extraction techniques, among others. Often, undersampling is helpful, but it can lead to a loss of information, affecting the performance of the classification model [19]. Unlike undersampling, oversampling methods, while prioritizing the integrity of information, do not remove a significant number of class samples during the synthesis process. This prevents the loss of essential class information that may occur in undersampling [20]. Among the many methods of oversampling, the Synthetic Minority Oversampling Technique (SMOTE) can take into account the samples of the original dataset to generate new data samples and has thus has been widely popularized [21,22]. Özdemir et al. used SMOTE-CNN to classify unbalanced hyperspectral images [23]. Liang et al. used SMOTE combined with XGBoost modeling to detect corn collapse [24]. In this study, to ensure that no information was missing from our study sample set as well as to address the small sample size and data imbalance in the peanut southern blight canopy spectra, we used SMOTE to synthesize new sample data.

The process of plants being subjected to pest and disease stress can be observed through factors such as tissue color, growth density, leaf shape, and more [25,26]. For optical remote sensing, a large number of studies have analyzed the spectral response under stress to diagnose the health status of plants [27,28]. Previous studies indicate that the spectral characteristics of peanut southern blight-affected leaves exhibit significant differences in the red edge and near-infrared regions [29]. Vegetation indices and mathematical transformations can be employed to assess the severity of disease under various growth conditions [30]. For instance, developing new spectral combinations based on the specificity of pests and diseases can be employed for monitoring *Ips typographus* using vegetation indices [31]. However, passive canopy spectral reflectance is susceptible to various factors such as solar angle, lighting conditions, canopy background, and shadows, which can impact the quality of the data [32]. Relying solely on raw spectral data for feature extraction is insufficient for disease detection. To enhance spectral detail and eliminate noise effects caused by factors like soil background, various mathematical transformation methods have been proposed by previous researchers, including CWT [33], SMC [34], SNV [35], differentiation [36], and others. Fractional-Order Differentiation (FOD), in contrast to integer-order, employs a more flexible order, reducing baseline drift and suppressing background noise. It has been widely utilized by many scholars for its advantages in better separating overlapping reflection peaks. Its applications include predicting soil characteristics [37], monitoring diseases [38], and tracking plant biochemical parameters [39]. FOD, built upon a small differential step size, further explores gradient information that is not inherent in integer-order differentiation, maximizing its retention throughout the process. In addition to data preprocessing, the selection of an appropriate model also significantly influences the monitoring of diseases. In the field of remote sensing for disease and pest monitoring, it is primarily divided into two major categories: machine learning and statistical models [40], machine learning models such as SVM, KNN, CNN, etc. have significantly alleviated the constraints of temporal and spatial dimensions, gaining widespread popularity for their excellent training errors and stronger generalization capabilities [41]. In particular, the

1D-CNN model excels at extracting "hidden" features from spectra and has been widely adopted in numerous studies. However, these studies often require large datasets for training [42]. For small datasets, can we overcome the limitations by utilizing synthetic datasets generated through techniques like SMOTE for model training? This approach aims to achieve optimal performance while validating against actual measured data, thereby overcoming data constraints—an aspect that has been seldom reported by scholars.

While ensuring the maximization of spectral information and enhancing the accuracy of the peanut southern blight mold detection model, addressing the challenges of data imbalance and insufficient sample size becomes increasingly intricate. Is it possible to employ SMOTE-FOD to synthesize new sample data, utilizing small differential steps to amplify differences between spectra? To our knowledge, previous studies on disease monitoring have not taken into account the premise of maximizing information, presenting a new challenge for us. The main objective of this study is to assess whether SMOTE-FOD-1D-CNN can enhance the detection performance of different severity levels of peanut southern blight. Specifically, we aim to address the following key questions: (1) Can SMOTE generate new sample data to resolve the issues of data imbalance and insufficient sample size while retaining maximum sample information? (2) Can the small-step FOD enhance spectral information and, in turn, elevate the potential of the constructed peanut southern blight detection model?

## 2. Materials and Methods

Experiment 1 was conducted in Zhengyang County, Zhumadian City, Henan Province, China, in the year 2022 (32°60′ N, 114°38′ E). The experimental area covered 4560 m$^2$, with planting initiated on 24 June 2022. During this period, the average temperature was 28 °C. The variety was the southern blight medium-resistant peanut variety Yuhua 37. Plant spacing was set at 35 cm, row spacing at 45 cm, and the experiment concluded on 6 September 2022. A total of 610 peanut canopy spectral data samples were collected, excluding 20 data samples where spectral measurements failed due to improper handling during the measurement process. The distribution of spectral samples for each category is as follows: 150 for healthy, 110 for mild, 120 for moderate, and 210 for severe.

Experiment 2 was conducted in Yanjin County, Xinxiang City, Henan Province, China, in September of the same year (35°15′57″ N, 114°11′8″ E). The climate was mild. Peanuts were a large-scale crop planted in the county with relatively uniform varieties, and standard farmland with a highly intensive planting density was selected. A total of 290 spectral samples representing various severity levels were collected in the experimental area, as illustrated in Figure 1.

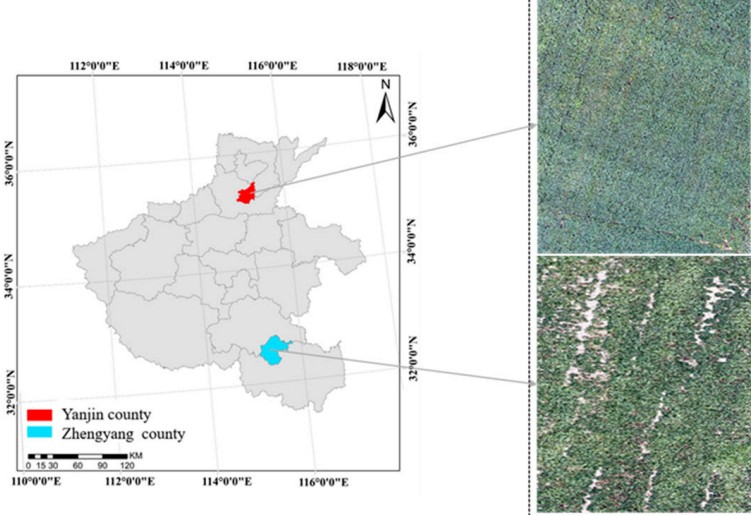

**Figure 1.** Overview map of the research area for peanut southern blight.

*2.1. Canopy-Spectral Data Acquisition*

The experiment utilized the ASD Field Spec3 spectrometer, with a spectral range of 350 to 2500 nm and spectral sampling intervals of 1.4 nm (350–1000 nm) and 2 nm (1001–2500 nm). The probe's field of view was approximately 25°, and the probe was positioned approximately 50 cm from the canopy during sampling. Measurements were conducted between 10:00 and 14:00 Beijing time under clear and cloudless weather conditions. Due to the dense planting of peanuts, we selected typical diseased plants for spectral data collection, with a measured area of approximately 0.1 m². During the measurement process, the probe collected spectral data from the canopy of peanut plants in a downward, vertical direction. And for each sampled point, radiometric calibration was performed using a BaSo4 standard whiteboard. Furthermore, due to the susceptibility of hyperspectral data to noise from factors such as photonic, atmospheric, and water vapor influences, this study removed the original spectra affected by noise and sidebands. The spectral range retained was from 600 to 1350 nm. Reflectance values were calculated using Formula (1).

$$R = \frac{L_t}{L_b} \times R_b \tag{1}$$

In Formula (1), where $R_b$ is the reflectance of the standard whiteboard, $L_t$ is the canopy radiance, $L_b$ is the whiteboard radiance, and $R$ is the calculated reflectance.

*2.2. Data Analysis Methods*

2.2.1. Investigation of Peanut Southern Blight Severity

A field investigation was conducted to assess the severity of peanut southern blight. Data acquisition of typical peanut plants in sampled locations based on previous studies on peanut genetic diversity and phenotype was guided by plant protection experts. Plants with a healthy root base, devoid of mycelium, and showing no symptoms on leaves were labeled as "Healthy". Plants with wilting in both leaves and stem bases affecting less than 1/3 of the entire plant were marked as "Mild". Plants with wilting in both leaves and stem bases affecting more than one-third but less than 2/3 of the entire plant were marked as "Moderate". Plants with wilting in both leaves and stem bases affecting more than 2/3 and the appearance of brown fungal sclerotia at the base were marked as "Severe". The four categories, including healthy peanuts and peanuts affected by southern blight, are represented by numerical values 1, 2, 3, and 4, corresponding to mild, moderate, and severe conditions, as illustrated in Figure 2.

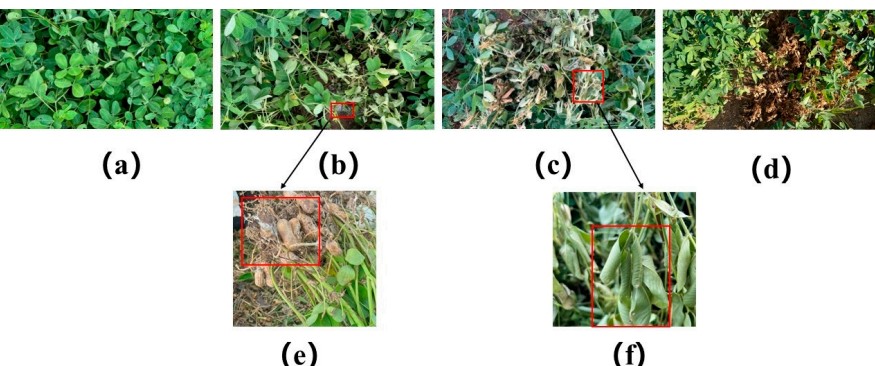

**Figure 2.** Field survey display of peanut southern blight. (**a**) Healthy plants, (**b**) plants with mild disease, (**c**) plants with moderate disease, (**d**) plants with severe disease. (**e**) Mycelium at the base of plants with mild disease, (**f**) leaf symptoms in plants with moderate disease.

2.2.2. Fractional-Order Differential

FOD theory extends the order of differentiation beyond integer values, allowing for arbitrary orders [43]. As a fundamental mathematical operation, FOD enables a more comprehensive analysis of signals [44]. The main forms include Riemann-Liouville (R-L),

Caputo, Weil, Caputo, and Grünwald-Letnikov (G-L). In this study, the discrete Grünwald-Letnikov (G-L) form is employed, which is suitable for small step sizes and computationally efficient. The specific formula is as follows in Equations (2) and (3):

$$D^\alpha f(x) = \lim_{h \to 0} \frac{1}{h^\alpha} \sum_{j=0}^{(t-\alpha)/h} (-1)^m \frac{\Gamma(\alpha+1)}{j!\Gamma(\alpha-j+1)} f(x-jh) \tag{2}$$

where $\alpha$ is the fractional order, $h$ is the step size, $t$ and $j$ are the upper and lower limits of differentiation, and $\Gamma$ is the gamma function.

$$\Gamma(\beta) = \int_0^\infty e^{-t} t^{\beta-1} dt = (\beta-1)! \tag{3}$$

The gamma function, also known as the generalized factorial, is commonly used in the definition of fractional-order differentials.

When $f(x)$ is a one-dimensional spectrum with a sampling interval of $1 (h = 1)$, and $t$ and $\alpha$ represent wavelength intervals, $(t-\alpha)/h = t - \alpha = n$. According to the above Equations (2) and (3), the following expression for fractional-order differentiation can be derived (Equation (4)):

$$\frac{d^v f(\lambda)}{d\lambda^v} \approx f(\lambda) + (-v)f(\lambda-1) + \frac{(-v)(-v-1)}{2}f(\lambda-2) + \cdots$$
$$\cdots + \frac{\Gamma(-v+1)}{(n)!\Gamma(-v+n+1)}f(\lambda-n) \tag{4}$$

When $v = 0$, the 0th-order differentiation of $f(x)$ is the function itself, and when $v = 1$, it represents the first-order differentiation.

### 2.2.3. 1D-CNN

The study employed a one-dimensional convolutional neural network (1D-CNN) to build the model [42]. Using peanut canopy spectra of different severity levels as input layer data, the convolutional layer extracts hidden features from the input layer. The feature dimensions are then reduced through a pooling layer. The flattened vector is then input into a fully connected layer using the ReLU activation function. The features extracted are mapped to the required size, representing the four severity levels of peanut southern blight in this study. The Adam gradient descent algorithm was employed with a maximum training iteration of 250, a batch size of 64, and an initial learning rate of 0.01. The 1D-CNN model was implemented using Matlab 2022a and NVIDIA RTX3060. The specific network parameters are detailed in Table 1.

**Table 1.** Parameters used in the 1D-CNN model.

| Stratification | Type | Output Features | Weight |
|:---:|:---:|:---:|:---:|
| 1 | Input Layer | $37 \times 1 \times 1$ | - |
| 2 | Convolutional Laye | $37 \times 1 \times 8$ | $2 \times 1 \times 1 \times 8$ |
| 3 | Batch Normalization Layer | $37 \times 1 \times 8$ | $1 \times 1 \times 8$ |
| 4 | ReLU | $37 \times 1 \times 8$ | - |
| 5 | Max Pooling Layer | $18 \times 1 \times 8$ | - |
| 6 | Convolutional Layer | $18 \times 1 \times 16$ | $2 \times 1 \times 8 \times 16$ |
| 7 | Batch Normalization | $18 \times 1 \times 16$ | $1 \times 1 \times 16$ |
| 8 | ReLU | $18 \times 1 \times 16$ | - |
| 9 | Fully Connected Layer | $1 \times 1 \times 4$ | $4 \times 288$ |
| 10 | Softmax | $1 \times 1 \times 4$ | - |
| 11 | Output Layer | $1 \times 1 \times 4$ | - |

### 2.2.4. SMOTE Algorithm

SMOTE (Synthetic Minority Oversampling Technique) is a method of data sampling that utilizes existing samples based on linear interpolation to generate new synthetic sample

data at random distances from the K-nearest neighbors [45]. The specific process is as follows: First, a minority sample *a* is selected from the original data. Then, the Euclidean distance between *a* and the remaining minority samples in the feature space is calculated to find the K-nearest neighbors. Based on the determined K-nearest neighbors, the SMOTE algorithm is used to select a random sample *b*, and finally, a new synthetic sample $\grave{a}$ is generated according to Equation (5).

$$\grave{a} \ = \ a + rand(0,1)|a - b| \tag{5}$$

The SMOTE algorithm has historically been utilized to address imbalances in minority classes within datasets. In recent years, as the efficacy of SMOTE's excellent oversampling strategy has been proven, an increasing amount of research suggests its applicability for synthetic sample generation. In this study, to determine the optimal number of K-neighbors, we set *K* = 1, 2, 3.

### 2.2.5. ReliefF Arithmetic

High spectral data has a small interval, a large quantity, contains a significant amount of noise and redundant features, and exhibits high multicollinearity among adjacent bands [46]. Blindly using all features will only increase meaningless computational load and decrease the model's generalization [47]. The ReliefF algorithm is one of the most commonly used methods for feature weight selection. Due to its simple calculation and high efficiency, it is widely used for the selection of multidimensional features [48]. In this study, the ReliefF algorithm was employed for rapid selection of features sensitive to peanut southern blight. The specific formula is as follows:

$$W^i(n) \ = \ W^{i-1}(n) - \sum_{j=1}^{k} \frac{(X_i(n) - H_j(n))^2}{mk} + \sum_{j=1}^{k} \frac{(X_i(n) - M_j(n))^2}{mk} \tag{6}$$

where *m* is the number of sampling times, *k* is the number of nearest neighbors, *M* is the selected number of features, $W^i(n)$ is the *n* weight factors updated for the *i*, $X_i(n)$ is the *n* feature index of the *i* random sample, $H_j(n)$ is the *n* feature of the *j* class's nearest sample, and $M_j(n)$ is the *n* feature index of the nearest samples of different classes for $X_i(n)$.

### 2.2.6. SVM and KNN Models

The Support Vector Machine (SVM) model seeks an optimal classification hyperplane in a high-dimensional space to separate different class samples with minimal error. SVM is commonly used for regression and classification problems. In this study, the kernel function is set to RBF [49]. The K-Nearest Neighbors (KNN) classification is based on the *K* most similar samples from the training set, using Euclidean distance for measurement. These samples vote on the attributes, and the final values are assigned to the object to be classified [50].

### 2.3. Model Accuracy Evaluation Metrics

In this study, we used the SMOTE-synthesized sample set for model training, constructed separate 1D-CNN, SVM, and KNN models, and evaluated the models' accuracy using 590 measured samples for validation. Additionally, to demonstrate the good generalization of the models, we tested them using 290 data samples from Yuxian County, Xinxiang City, Henan Province. The overall classification accuracy and Kappa coefficient were calculated based on the confusion matrix of the models, as shown in Equations (7) and (8).

$$\text{OA} \ = \ \frac{\left( \sum_{k=1}^{N} akk \right)}{n} \tag{7}$$

where $N$ is the number of classes, $n$ is the total number of classifications, and $akk$ is the number of correct classifications.

$$\text{Kappa} = \frac{N\sum_{i=1}^{m} x_{ii} - \sum_{k=1}^{m}\left(\sum_{i=1}^{m} x_{ij}\sum_{j=1}^{m} x_{ij}\right)}{N^2 - \sum_{k=1}^{m}\left(\sum_{i=1}^{m} x_{ij}\sum_{j=1}^{m} x_{ij}\right)} \tag{8}$$

where $N$ is the total, $x_{ii}$ is the diagonal element of the confusion matrix, and $x_{ij}$ is each element of the confusion matrix.

## 3. Results

### 3.1. Synthetic Data Generation

In this study, we used the SMOTE algorithm to synthesize the original data. To further analyze the accuracy of the synthetic data and the optimal number of nearest neighbors (K), we first fixed the generation multiplier at $n = 10$, with a focus on analyzing the similarity between synthetic data and real data for $K = 1, 2,$ and 3. To illustrate this situation more intuitively, we used PCA to plot the spatial distribution of the first three principal components, as shown in Figure 3. Compared to the original data, all of the generated synthetic data exhibit excellent spatial similarity.

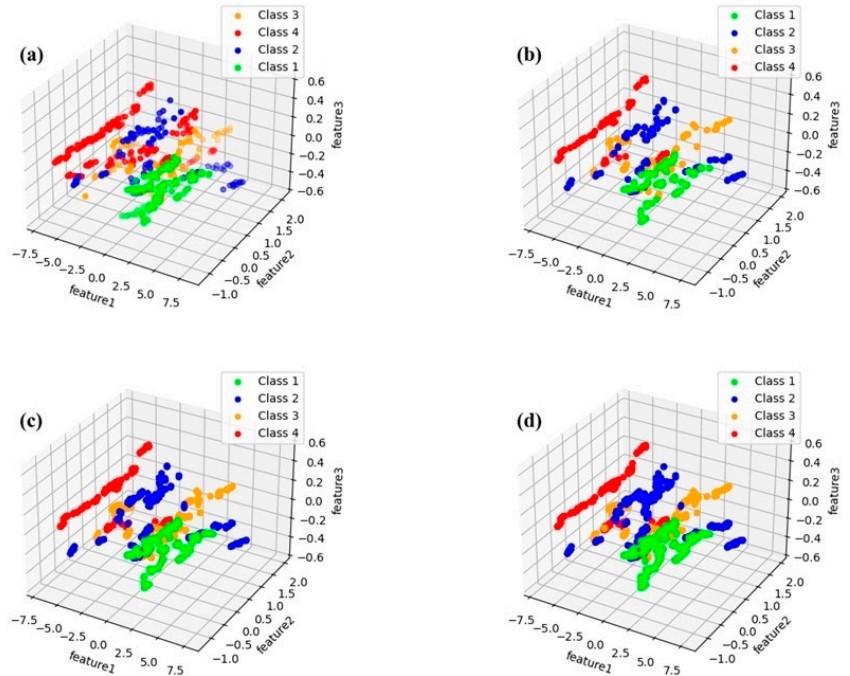

**Figure 3.** Spatial distribution maps of original data and synthetic samples for different values of $K$ based on PCA. (**a**) Original data. (**b**) Synthetic data for $K = 1$. (**c**) Synthetic data for $K = 2$. (**d**) Synthetic data for $K = 3$. The legend Class 1–4 represents the four categories in this study.

### 3.2. Features of Spectral Curves under Different Fractional Differentiation Orders

The spectral curves of peanut southern blight at different severities are distinct. After infection with peanut southern blight, the physiological and biochemical parameters of peanuts undergo significant changes, resulting in a decrease in reflectance in the red edge and near-infrared regions, causing a 'blue shift' phenomenon. Monitoring peanut southern blight solely based on the spectral response mechanism still poses challenges, as the canopy environment is always complex and susceptible to various factors. As shown in Figure 4a, spectral curves of different severities intersect and appear similar.

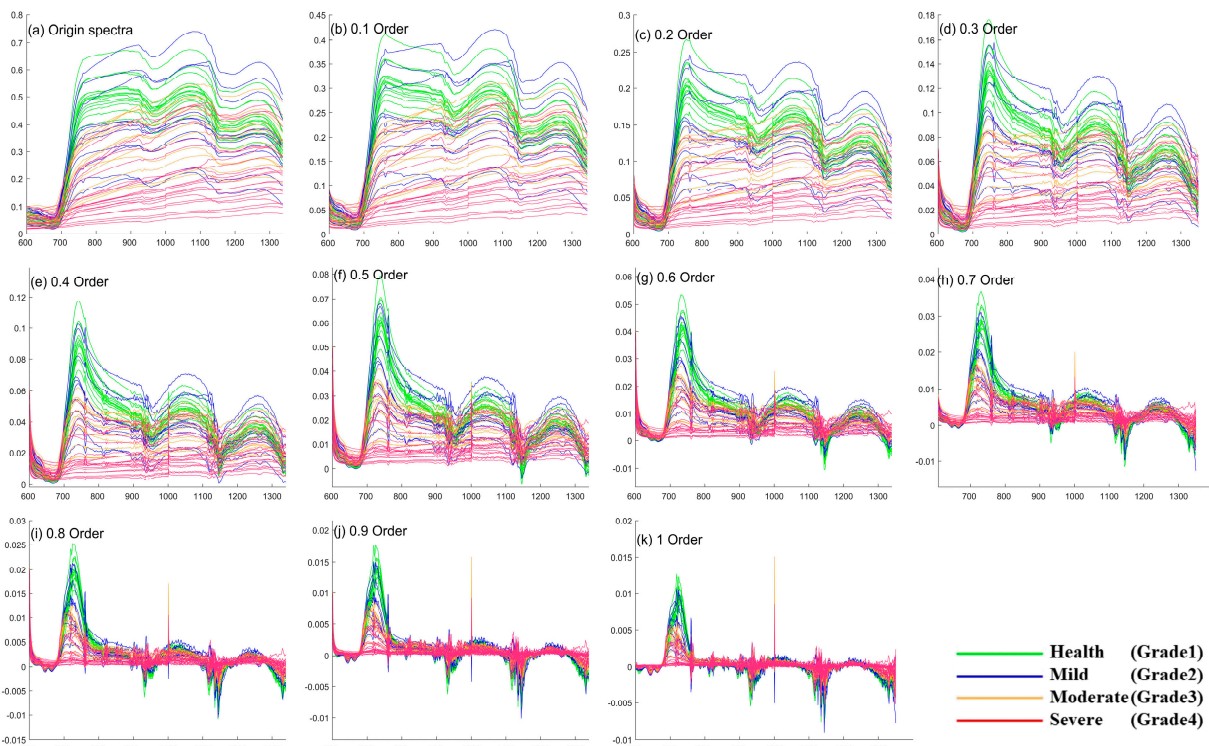

**Figure 4.** FOD spectral curves in the (0–1.0) fractional-order range.

To investigate the improvement in FOD on peanut southern blight spectra, we applied fractional-order differences to the original spectra with a step size of 0.1, obtaining spectral curves for different severities as shown in Figure 4a–k The original spectra are smooth, with two weak absorption features near 970 nm and 1200 nm influenced by internal leaf structures. Absorption valleys and reflection peaks increase with the FOD order, and at FOD = 0.5, reflectance becomes negative, with significant fluctuations in many curves. As shown in Figure 5a–j, after FOD = 1.1, this fluctuation trend gradually slows down, and after FOD = 1.4, the dimensional differences between various spectral curves gradually decrease, showing signs of overlap. At the same time, the spiky trend of the spectral curves deepens, and the entire spectrum becomes chaotic.

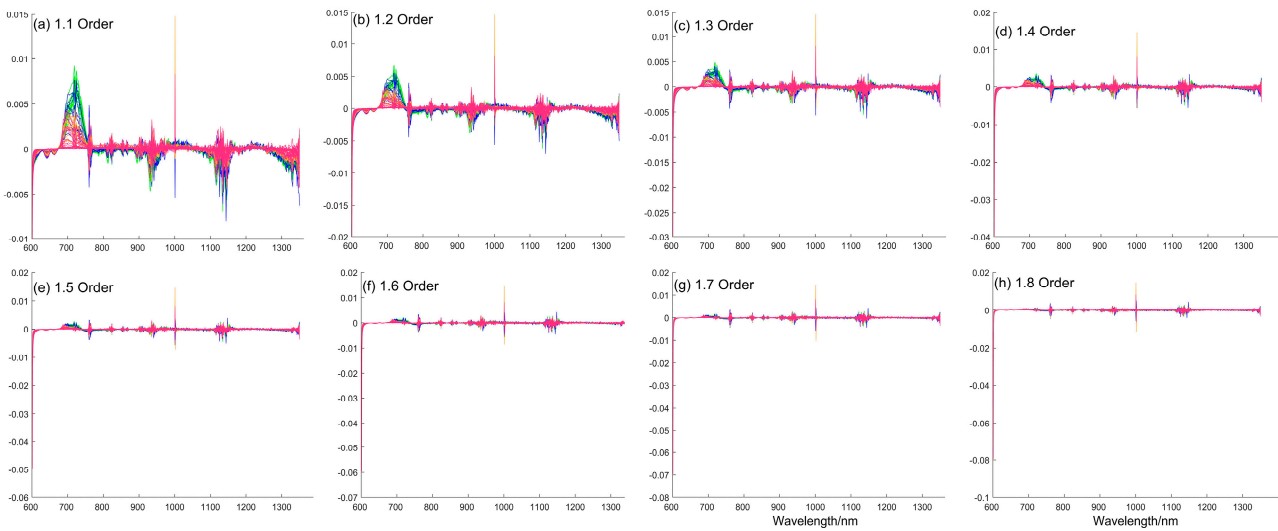

**Figure 5.** *Cont.*

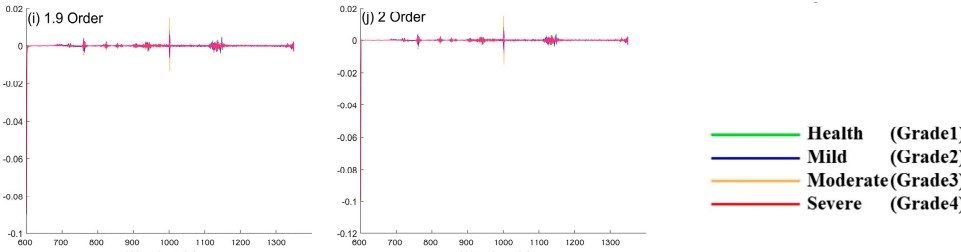

**Figure 5.** FOD spectral curves in the (1.1–2.0) fractional-order range.

### 3.3. Correlation between Disease Severity and Spectra

To assess whether FOD can improve the correlation between disease severity and spectral reflectance, we compared the correlation information between FOD spectra and disease severity at different orders with a step size of 0.1 (Figure 6). For the original spectra, at FOD = 0, a significant negative correlation was observed in the red edge, with the maximum correlation at 756 nm (R = −0.83). As the FOD order increased, the correlation coefficient changed. For instance, at FOD = 1.1, the correlation gradually became positive around 780 nm. Moreover, the spectra correlations based on FOD were generally better than the original spectra, indicating that using FOD could enhance the spectral details and increase the separability of disease severity.

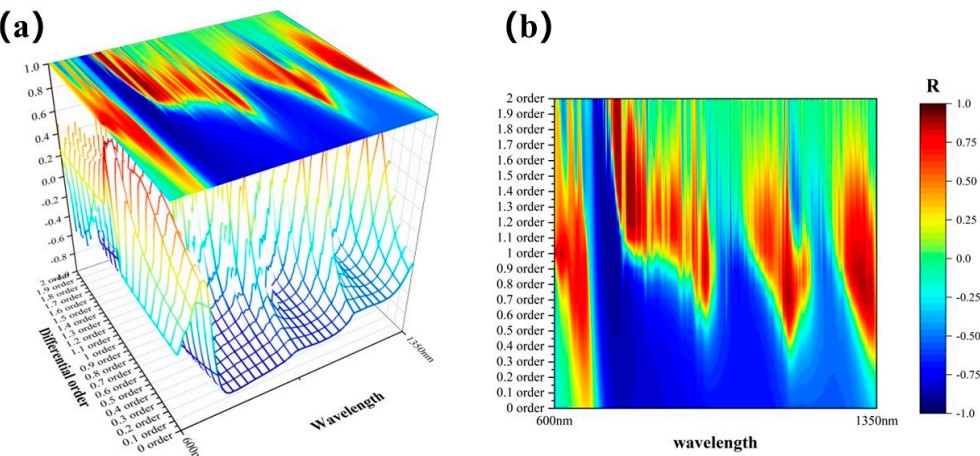

**Figure 6.** Correlation between FOD spectra and diseases at different orders. (**a**) 3D spectral curve correlation projection diagram. (**b**) 2D correlation diagram.

### 3.4. ReliefF Feature Selection Algorithm

Due to the fact that hyperspectral data consist of thousands of closely adjacent narrow bands with significant inter-band correlations, this study utilized the ReliefF algorithm for feature selection. The algorithm was applied to screen out the top 5% weighted features under different scenarios, including SMOTE synthesized data and actual measured data, with varying *K* values (1, 2, 3) and different FOD ranging from 0.1 to 2.0. The selected features were then employed in the construction of the disease detection model. The process and results are illustrated in Figure 7.

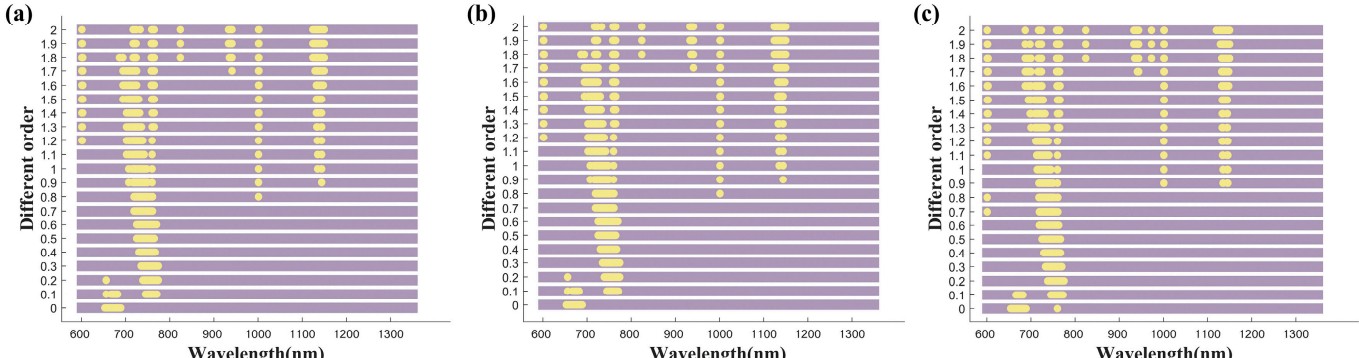

**Figure 7.** Illustration of feature selection using the ReliefF algorithm for the top 5% weighted features under different scenarios of FOD (0–2). The selected features are highlighted in yellow. Specifically, the features for synthesized data are represented when (**a**) $K = 1$, (**b**) $K = 2$, and (**c**) $K = 3$.

*3.5. Construction of Disease Detection Model Based on FOD Spectra*

3.5.1. Performance of Multiple Outputs in the 1D-CNN Model

In order to investigate the impact of different FODs on the detection of peanut southern blight and determine the optimal $K$ neighbors for the SMOTE algorithm, we applied the 1D-CNN model to spectra processed with 20 different FOD orders, as shown in Table 2. When $K = 1$ and FOD = 0.2, the overall accuracy (OA) was 88.81%, and the Kappa coefficient was 0.85, which was better than the performance with the original spectra (FOD = 0, OA = 86.78%, Kappa = 0.82). Furthermore, we analyzed the variation of overall accuracy with different values of $K$ for the entire range of FOD, as shown in Figure 8. We observed a decreasing trend in overall accuracy with increasing $K$ values, and the overall accuracy was generally higher when $K = 1$ compared to other values.

**Table 2.** Based on the 1D-CNN monitoring model of peanut southern blight under various orders of FOD.

| FOD Order | K = 1 | | K = 2 | | K = 3 | |
|---|---|---|---|---|---|---|
| | OA | Kappa | OA | Kappa | OA | Kappa |
| 0 | 86.78% | 0.82 | 87.29% | 0.83 | 87.12% | 0.83 |
| 0.1 | 86.10% | 0.81 | 86.10% | 0.81 | 85.93% | 0.81 |
| 0.2 | 88.81% | 0.85 | 85.76% | 0.81 | 87.46% | 0.83 |
| 0.3 | 87.46% | 0.83 | 87.12% | 0.83 | 86.44% | 0.82 |
| 0.4 | 86.44% | 0.82 | 86.95% | 0.82 | 85.93% | 0.81 |
| 0.5 | 86.61% | 0.82 | 86.27% | 0.81 | 86.44% | 0.82 |
| 0.6 | 86.78% | 0.82 | 86.44% | 0.82 | 85.93% | 0.81 |
| 0.7 | 86.44% | 0.82 | 84.41% | 0.79 | 86.95% | 0.82 |
| 0.8 | 85.42% | 0.8 | 84.24% | 0.79 | 82.71% | 0.77 |
| 0.9 | 85.25% | 0.8 | 83.05% | 0.77 | 83.56% | 0.78 |
| 1.0 | 84.07% | 0.79 | 82.03% | 0.76 | 83.22% | 0.77 |
| 1.1 | 83.90% | 0.78 | 85.08% | 0.8 | 83.73% | 0.78 |
| 1.2 | 83.22% | 0.77 | 82.37% | 0.76 | 82.71% | 0.77 |
| 1.3 | 84.07% | 0.79 | 83.22% | 0.77 | 84.41% | 0.79 |
| 1.4 | 83.39% | 0.78 | 81.69% | 0.75 | 81.02% | 0.74 |
| 1.5 | 82.54% | 0.77 | 81.36% | 0.75 | 83.56% | 0.78 |
| 1.6 | 82.20% | 0.76 | 80.00% | 0.73 | 81.02% | 0.74 |
| 1.7 | 79.83% | 0.73 | 80.51% | 0.73 | 80.17% | 0.73 |
| 1.8 | 77.29% | 0.69 | 77.29% | 0.69 | 75.25% | 0.66 |
| 1.9 | 74.07% | 0.65 | 71.86% | 0.61 | 72.03% | 0.62 |
| 2 | 73.73% | 0.64 | 70.85% | 0.6 | 71.69% | 0.62 |

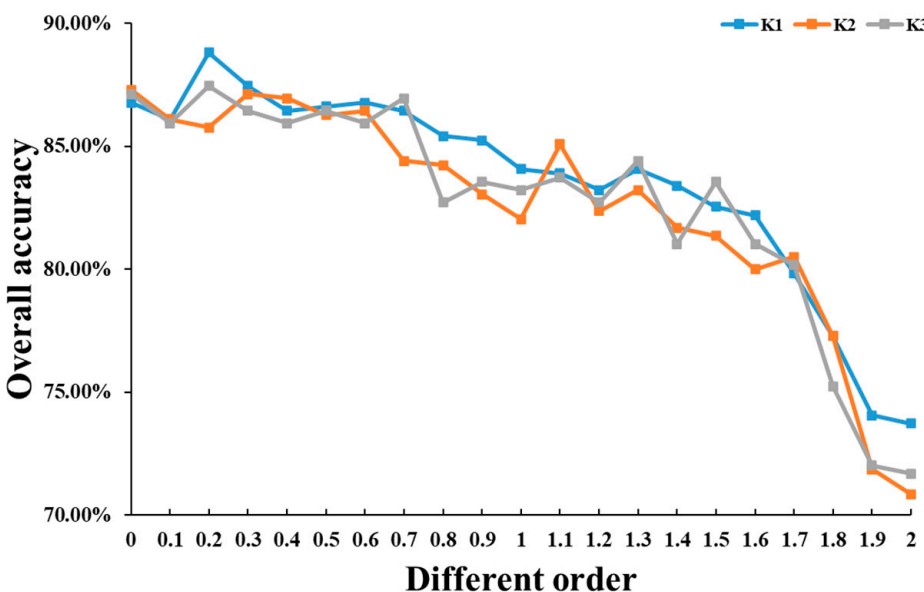

**Figure 8.** Overall accuracy of the 1D-CNN model under different K-nearest neighbors values.

### 3.5.2. Performance of Machine Learning Models with Multiple Outputs

To compare the performance of the models and determine the optimal FOD order, we further analyzed the performance of the SVM and KNN models with the best neighbor number $K = 1$ for each FOD. As shown in Table 3, when FOD is in the range of 0.2–0.3, the overall accuracy (OA) is better than other orders, consistent with the results of the 1D-CNN model. The best accuracy of the SVM model was FOD = 0.3 (OA = 86.61%, Kappa = 0.82), and the best accuracy of the KNN model was FOD = 0.2 (OA = 86.95%, Kappa = 0.83).

**Table 3.** Peanut southern blight monitoring model based on SVM and KNN under FOD.

| | Model ($K = 1$) | | | |
|---|---|---|---|---|
| **FOD Order** | **SVM** | | **KNN** | |
| | **OA (%)** | **Kappa** | **OA (%)** | **Kappa** |
| 0 | 72.20% | 0.63 | 84.75% | 0.80 |
| 0.1 | 70.34% | 0.60 | 84.24% | 0.79 |
| 0.2 | 84.07% | 0.79 | 86.95% | 0.83 |
| 0.3 | 86.61% | 0.82 | 85.76% | 0.81 |
| 0.4 | 83.90% | 0.78 | 80.68% | 0.74 |
| 0.5 | 79.32% | 0.72 | 75.76% | 0.68 |
| 0.6 | 74.41% | 0.66 | 71.53% | 0.62 |
| 0.7 | 75.59% | 0.68 | 72.88% | 0.64 |
| 0.8 | 75.42% | 0.67 | 72.20% | 0.63 |
| 0.9 | 77.29% | 0.69 | 72.54% | 0.63 |
| 1.0 | 78.64% | 0.71 | 70.68% | 0.60 |
| 1.1 | 77.46% | 0.70 | 73.22% | 0.64 |
| 1.2 | 76.10% | 0.68 | 73.73% | 0.65 |
| 1.3 | 75.93% | 0.68 | 71.53% | 0.61 |
| 1.4 | 74.92% | 0.66 | 67.46% | 0.55 |
| 1.5 | 72.37% | 0.63 | 64.24% | 0.51 |
| 1.6 | 70.68% | 0.60 | 59.83% | 0.46 |
| 1.7 | 67.12% | 0.55 | 54.24% | 0.38 |
| 1.8 | 63.22% | 0.49 | 51.53% | 0.35 |
| 1.9 | 60.00% | 0.45 | 49.83% | 0.33 |
| 2 | 57.80% | 0.42 | 49.83% | 0.32 |

3.5.3. Evaluation of Model Generalization Performance

To assess the robust generalizability of the constructed model, we tested it using 290 spectral samples from various regions with different severity levels in the same year. Additionally, mean processing was conducted prior to utilization. The results are shown in Table 4. It can be observed that under the optimal input parameters, the 1D-CNN model achieved good performance on the validation set, with an accuracy (OA) of 82.76% and a Kappa of 0.75, while the performance of the other machine learning models on the test set was not satisfactory.

**Table 4.** Accuracy assessment of different models on test data.

| Model Input Parameters | Model | Calibration | | Validation | |
|---|---|---|---|---|---|
| | | OA (%) | Kappa | OA (%) | Kappa |
| | 1D-CNN | 88.81% | 0.85 | 82.76% | 0.75 |
| $K = 1$, $n = 10$, FOD = 0.2 | SVM | 84.07% | 0.79 | 48.28% | 0.27 |
| | KNN | 86.95% | 0.83 | 65.52% | 0.49 |

**4. Discussion**

In this study, to monitor the severity of peanut southern blight in the canopy, we used the SMOTE algorithm to synthesize canopy spectral data for peanut southern blight. Furthermore, we employed FOD for in-depth analysis. The study is divided into three parts for discussion. Firstly, we analyzed the similarity between the synthesized data using the SMOTE algorithm and real data. We explored the impact of the optimal nearest neighbor number (K) and multiplier (n). Secondly, we compared and analyzed the 1D-CNN model and SVM model, investigating the optimal FOD order for peanut southern blight and further analyzing it with a step size of 0.01. Finally, we summarized the shortcomings of this study and provided future prospects.

*4.1. SMOTE Analysis of Synthetic Data*

The early stages of peanut southern blight differ from stem rot [51] and leaf spot diseases [52] in that there are no obvious pathogenic features on the plant canopy. Once the fungal hyphae, which have infected the plant internally, are subjected to continuous high-temperature and high-humidity conditions, they will rapidly spread [53]. The monitoring window for this early stage is very short, and our study did not involve continuous monitoring for an extended period, resulting in a limited number of early stage samples obtained.

Although data mining techniques have been widely promoted, we concur with Guo's viewpoint that traditional classification modeling for imbalanced datasets remains a pressing issue [21]. As mentioned by López and Weiss, when dealing with imbalanced sample sets, training machine learning standard classifiers may yield ideal results, but the minority sample set is still neglected. This could potentially lead to poor robustness of the model in the face of rare events and an increased likelihood of misidentifying minority class samples as noise, thereby impacting the overall model accuracy [54,55].

SMOTE does not require training any specific model; it generates new synthetic samples solely based on the spatial characteristics of the original data using the K-nearest neighbor approach [56,57]. The choice of the *K* value directly influences the reliability of the synthesized data, depending on the density or sparsity of the initial data distribution [58,59]. To illustrate the distribution of the synthetic data compared to the original data more intuitively, as shown in Figure 3, we analyzed the spatial features of the first three PCAs. The spatial features of the synthetic data are extremely similar to those of the original data. However, we do not assert that synthetic data guarantees good performance for such datasets; it merely satisfies visual consistency, as confirmed by Kristian [60]. Further analysis of the subsequent data is still needed.

On this basis, to determine the optimal values for *K* (number of nearest neighbors) and the synthesis multiplier n, we tested the data results for n = 10 with K1, K2, and K3 separately. As shown in Figure 8, we found that overall FOD differential accuracy is better when using K1 compared to the other values. Using a smaller k can generate a representative dataset for our data, and we agree with Ebrahimy et al., that further experiments are needed to evaluate other crops [61]. In addition, in recent years, the multiplier n for synthetic data has also been widely discussed [62]. In this study, we attempted to analyze the impact of different synthetic multipliers on the model to determine the optimal value for n. We used SVM, the least effective model among the three, for this analysis, and the results are shown in Figure 9. As the synthesis data multiplier n increases, the overall accuracy stabilizes after n = 10. As mentioned by Sun, blindly continuing to increase the synthesis multipliers n and k can undoubtedly increase computational load, which is often undesirable while maintaining optimal accuracy [20].

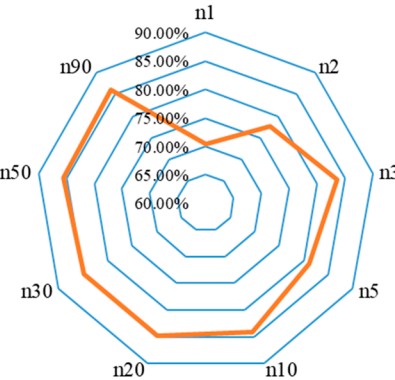

**Figure 9.** SVM model accuracy with different multiples (n) of synthetic data.

Although SMOTE can synthesize new sample data while ensuring the integrity of information, it is necessary to mention its disadvantage of generating noisy samples [15]. In addressing this drawback, previous researchers have proposed methods to identify and eliminate noise, such as improved SMOTE-IPF [63], SMOTE-Rknn [64], etc. However, in this study, we still aim to retain this noise because we understand that some scholars, in radiative transfer models, add varying degrees of Gaussian noise to simulated datasets to minimize overfitting issues, aligning with their approach [65]. Moreover, the uncertainty of biochemical parameter retrieval can be influenced by the quality of the data, as crown reflectance may vary [66]. Therefore, as shown in Figure 10, we magnified the differences between the synthetic sample dataset and the measured data, allowing for a visual observation of the slight noise issue in the synthetic samples.

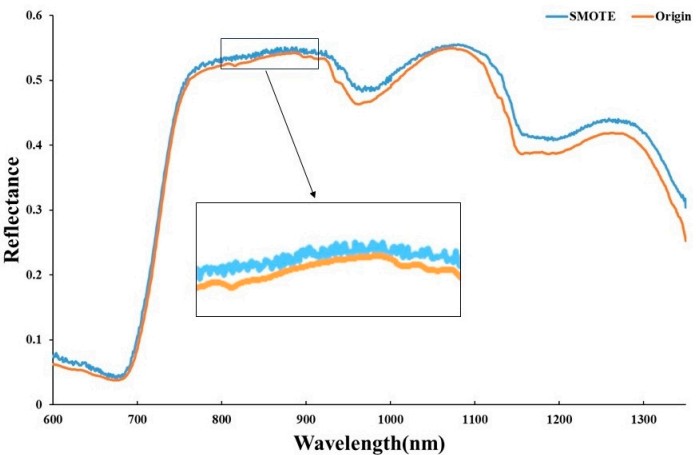

**Figure 10.** Spectral curves of SMOTE synthetic data and actual measured data.

*4.2. Analysis of Various Orders of FOD*

Integer-order differentiation is just a special case of fractional-order differentiation. Previous studies have indicated that first-order differentiation is effective in eliminating the influence of background noise, while second-order differentiation can mitigate baseline drift, thereby enhancing analytical accuracy [67]. However, some researchers have proposed that higher-order integer differentiations may introduce noise and compromise the integrity of information [38]. Fractional-order differentiation, while encompassing the meaning of integer-order differentiation, provides a more flexible choice of orders [68,69]. Fractional-order differentials are closely related to full-band reflectance, where the reflectance of each point is influenced by different fractional-order differentials with varying weights. The closer the point, the larger the weight, and hence, the greater the influence of fractional-order differentials. This is what Zhang referred to as the memory and non-locality of fractional-order differentials [70]. For disease spectra, peanut southern blight causes rapid yellowing and wilting of the entire plant, leading to significant changes in the canopy spectra, making the spectral information more complex. Therefore, we attempt to use fractional-order differentials to further amplify subtle spectral differences and explore the capability of this method for monitoring peanut southern blight.

As shown in Figure 4, a fractional-order differential transformation with a step size of 0.1 is applied to the original spectra, and the overall spectral shape exhibits a significant changing trend. Before FOD = 0.9, with the increase in order, the spectral differences are magnified, leading to many distinctive absorption valleys and reflection peaks. However, at higher orders, this unique phenomenon is masked, and the separability of various spectral curves gradually decreases. Zhang suggested that this phenomenon could be explained using the G-L mathematical theory. When the sampling step is smaller than the width of the peaks and valleys, spectral differences are magnified. The noise phenomenon arises because the calculation of FOD introduces high-frequency noise when dealing with short-interval peaks and valleys in the spectrum [38]. This phenomenon can also be observed in Figure 6. Additionally, we observed that while FOD enhances the correlation between the spectrum and peanut southern blight, it also leads to certain originally positive correlations in specific bands becoming negative, and vice versa. The original bands exhibit a significant negative correlation near the red edge, and with the increase in FOD, it gradually turns positive around FOD = 1.1. This conclusion has been supported by various studies; Jiang et al. suggested that it is due to the capture of long-term memory and non-local features in the data by FOD [37], while Kilbas et al. proposed that it might be caused by nonlinear relationships between variables [69].

In order to further investigate the impact of FOD on the monitoring of the severity of peanut southern blight, we used 1D-CNN, SVM, and KNN models for evaluation. As shown in Tables 2 and 4, with the increase in FOD order, the overall accuracy shows an upward and then downward trend. FOD = 0.2 is the optimal order, and all three models have high evaluation metrics. The 1D-CNN model is optimal when $K = 1$ and n = 10 (validation set OA = 88.81%, Kappa = 0.85; test set OA = 82.76%, Kappa = 0.75). Analyzing smaller step sizes on top of limited orders to determine if it can increase model accuracy is an aspect that has not been explored in previous research. Next, we conducted a detailed analysis of the impact of FOD on peanut southern blight detection with a step size of 0.01. The accuracy using SVM and KNN models is shown in Figure 11. There is not a significant improvement in accuracy within the range of FOD from 0.21 to 0.29. The ability to decompose spectral information using FOD with a small step size has reached saturation.

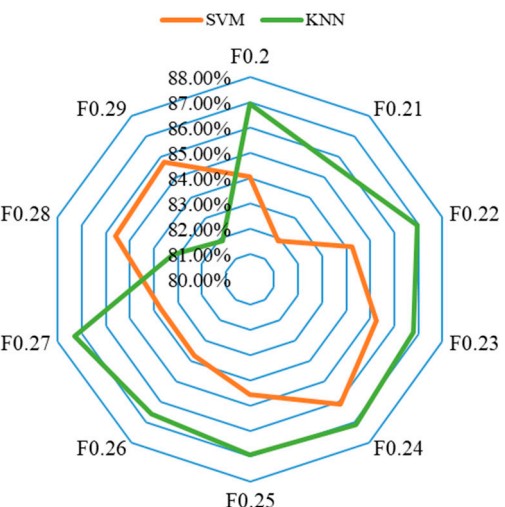

**Figure 11.** Peanut southern blight severity detection using different models with FOD (0.2–0.29) and a step size of 0.01.

### 4.3. Limitations and Future Work

On one hand, while our study demonstrates that using SMOTE to generate training samples, further amplifying spectral differences between different severities with FOD, and employing a 1D-CNN model can effectively monitor peanut southern blight, investigations at different stages of peanut southern blight were not conducted. The main reason for this is the transitional continental monsoon climate in Henan Province, transitioning from a northern subtropical zone to a warm temperate zone. The region experiences a high probability of continuous high temperatures and humidity in July and August [71], providing favorable external conditions for the occurrence of peanut southern blight. Early monitoring in the growing season is hindered by the influence of peanut growth density, as the pathogenic hyphae do not erupt extensively in the early stages. On the other hand, due to the highly complex nature of field environments, diseased plants may be subjected to various types of stressors, such as drought, flooding, nutrient deficiencies, and more. Additionally, for the plants themselves, they may be affected by more than one disease during their growth process; the occurrence of multiple diseases concurrently is something that we have not thoroughly considered.

Our future work will extend beyond the scope of Henan Province. We aim to acquire data on peanut southern blight from multiple provinces and varieties to enhance the model. This expansion is crucial for early disease detection, improving prevention and control efficiency, and cultivating disease-resistant varieties. At the same time, we need to further refine our experimental design, make fuller use of experimental resources, systematically control variables, and expand future work to assess the effects of various stressors.

### 5. Conclusions

This study aimed to enhance the ability to monitor the severity of peanut southern blight by using the SMOTE-FOD-1D-CNN model. A comparison with the SVM and KNN models was conducted, and the optimal model was determined to be 1D-CNN (validation OA = 88.81%, Kappa = 0.85; test OA = 82.76%, Kappa = 0.75). The main conclusions are as follows: (1) SMOTE effectively addresses the issue of imbalanced and insufficient sample data by preserving the maximum amount of sample information and generating new sample data. (2) Small-step FOD shows potential for enhancing spectral information and improving the performance of the constructed peanut southern blight monitoring model.

**Author Contributions:** H.S.: investigation, methodology, validation, writing—original draft, visualization. L.Z.: investigation, methodology. M.S.: investigation, methodology. J.Z.: methodology, visualization. Z.F.: methodology, visualization. H.F.: methodology, visualization. X.S.: investigation, methodology. J.Y.: investigation, methodology, validation. W.G.: investigation, methodology, validation, writing—review and editing. All authors have read and agreed to the published version of the manuscript.

**Funding:** This work was supported by the Henan Provincial Science and Technology Major Project (NO.221100110100); National Natural Science Foundation of China, grant number (NO.32271993); The Joint Fund of Science and Technology Research Development program (Cultivation project of preponderant discipline) of Henan Province, China (NO.222301420114).

**Institutional Review Board Statement:** Not applicable.

**Data Availability Statement:** The data presented in this paper are part of an ongoing study and the dataset is difficult to access; permission from the corresponding author is required to access the dataset.

**Conflicts of Interest:** The authors declare that they have no known competing financial interests or personal relationships that could have appeared to influence the research reported in this study.

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
