# Peer review of "Estimation of Peanut Southern Blight Severity in Hyperspectral Data Using the Synthetic Minority Oversampling Technique and Fractional-Order Differentiation"

_agriculture, doi:10.3390/agriculture14030476_

Round 1

Reviewer 1 Report

Comments and Suggestions for Authors

The authors addressed a critical, practical point of view, the problem of monitoring the degree of peanut southern blight using a non-invasive spectral method. It is also a scientific challenge related to developing an appropriate model that can be used to monitor the development of southern peanut blight quickly.

The usefulness of the model selected by the authors can be assessed using data from continuous and long-term monitoring of crops of various varieties of peanuts from different provinces.

Comments on the Quality of English Language

There are minor stylistic errors. I suggest that a native speaker reviews the study.

Author Response

Dear expert,

Thank you very much for your guidance and feedback. Below is my response to the suggested modifications.

  • Reviewer #1: There are minor stylistic errors. I suggest that a native speaker reviews the study.

    Response:

Thanks to the expert's comments. To address this issue, we invited an English language expert to revise the entire text a second time.

Reviewer 2 Report

Comments and Suggestions for Authors

The manuscript describes a method for measuring Peanut southern blight severity based on hyperspectral data and machine learning techniques. The manuscript is well written, the experiments are well designed and, overall, the study is quite thorough. More specifically:
- The introduction is well written and provides an adequate contextualization for the remainder of the manuscript. The study’s contributions are clearly stated, especially regarding the research gaps it tries to address. Well done!
- The “Materials and Methods” section is also well written. My only suggestion would be to better explain the difference between “spectral data points” and “spectral samples”, because both terms and associated amounts appear several times and it is difficult to know exactly what they refer to.
- The “Results” and “Discussion” sections are thorough and bring many useful insights. I do not have any further suggestions.

Comments on the Quality of English Language

Apart from a few minor language issues which can be easily corrected, the English used throughout the text is adequate and easy to understand.

Author Response

Dear expert,

Thank you very much for your guidance and feedback. Below is my response to the suggested modifications.

  1. Reviewer #2: The manuscript describes a method for measuring Peanut southern blight severity based on hyperspectral data and machine learning techniques. The manuscript is well written, the experiments are well designed and, overall, the study is quite thorough. More specifically:
    - The introduction is well written and provides an adequate contextualization for the remainder of the manuscript. The study’s contributions are clearly stated, especially regarding the research gaps it tries to address. Well done!
    - The “Materials and Methods” section is also well written. My only suggestion would be to better explain the difference between “spectral data points” and “spectral samples”, because both terms and associated amounts appear several times and it is difficult to know exactly what they refer to.
    - The “Results” and “Discussion” sections are thorough and bring many useful insights. I do not have any further suggestions.
    Response:Thanks to the expert's comments. For this text, we have made consistent changes throughout the text to use "spectral samples" instead. And use red font to mark the modified parts. Specifically:
  • Replaced "spectral data points" with "spectral samples" in line 290.
  • Substituted "spectra data" with "spectral samples" in line 356. Additionally, we added "Additionally, mean processing was conducted prior to utilization." at the end of line 357.

Reviewer 3 Report

Comments and Suggestions for Authors

The manuscript entitled „Estimation of Peanut southern blight Severity in Hyperspectral Data Using Synthetic Minority Oversampling Technique and Fractional Order Differentiation” presents interesting study on remote sensing and application of various data analysis methods for disease detection.

Please select the Type of the Paper. In this case it is “article” as I assume.

“southern blight” in the title of the manuscript should be written using capital letters.

In the study ASD Field Spec3 spectrometer was used for data acquisition. The sensor measures reflectance of very small area. How big was area which was measured? Please write more information about these measurements. Reflectance of leaves or other parts of the plant were measured?

The main problem in such studies in detection of specific disease symptoms is not discrimination between healthy and infected plants but discrimination between plants with specific stresses, for example discrimination between two or more diseases or discrimination between disease symptoms and water stressed plants. The study is limited to only one disease. In such case we can use RGB data and distinguish quite simply between healthy plants vs. infected. It is quite easy issue. Could you prove that spectral reflectance is this disease specific?

In the discussion I suggest to add information which confirms that obtained patterns of the reflectance is disease specific or not. In most studies discrimination based on spectral data is limited to only one disease, but in agricultural practice plants can be infected by more than one disease or stressed by many different factors, including water stress, nutrients deficiency,…

In “Limitation and future work” authors wrote: “We aim to acquire data on peanut southern blight from multiple provinces and varieties to enhance the model”. In my opinion the future work should be extended to evaluation of different stress types (different diseases symptoms, water stress, nutrient deficiency symptoms, …) not only limited to one disease.

Author Response

Dear expert,

Thank you very much for your guidance and feedback. Below is my response to the suggested modifications.

  1. Reviewer #3: Please select the Type of the Paper. In this case it is “article” as I assume.

Response:Thanks to the expert's comments. We determined the type of paper to be an article.

  1. Reviewer #3: “southern blight” in the title of the manuscript should be written using capital letters.

Response:Thanks to the expert's comments. We modify the phrase 'southern blight' in the title to uppercase letters.

  1. Reviewer #3: In the study ASD Field Spec3 spectrometer was used for data acquisition. The sensor measures reflectance of very small area. How big was area which was measured? Please write more information about these measurements. Reflectance of leaves or other parts of the plant were measured?

Response:Thanks to the expert's comments. In response to this issue, we have supplemented the measurement information in section 2.1.1. The specific details are as follows: Due to the dense planting of peanuts, we selected typical diseased plants for spectral data collection, with a measured area of approximately 0.1. During the measurement process, the probe collected spectral data from the canopy of peanut plants in a downward vertical direction.

  1. Reviewer #3: The main problem in such studies in detection of specific disease symptoms is not discrimination between healthy and infected plants but discrimination between plants with specific stresses, for example discrimination between two or more diseases or discrimination between disease symptoms and water stressed plants. The study is limited to only one disease. In such case we can use RGB data and distinguish quite simply between healthy plants vs. infected. It is quite easy issue. Could you prove that spectral reflectance is this disease specific?

Response:Thanks to the expert's comments. We fully agree with your point, but currently, research on peanut southern blight remains limited. On one hand, as mentioned in our main text, early-stage peanut southern blight differs from common peanut diseases such as stem rot and leaf spot; it lacks particularly distinct pathological features in the canopy. Therefore, during field collection, it requires a significant amount of time to locate typical samples. On the other hand, our other study involves using drones to monitor peanut white vein disease. Currently, our findings also indicate that using RGB data alone to monitor the occurrence of early Peanut southern blight remains challenging. We are aware of the complexity of field environments, where phenomena like same-spectrum different objects do occur. Our preliminary results from the greenhouse inoculation experiment published in the Agriculture journal in 2023 suggest that the spectral response on the leaves possesses this characteristic. The inadequacy of research on the effects of multiple stressors is indeed a limitation of our experiments. Moving forward, we will redesign our experimental approach based on your valuable feedback.

  1. Reviewer #3: In the discussion I suggest to add information which confirms that obtained patterns of the reflectance is disease specific or not. In most studies discrimination based on spectral data is limited to only one disease, but in agricultural practice plants can be infected by more than one disease or stressed by many different factors, including water stress, nutrients deficiency,…

Response:Thanks to the expert's comments. We agree with your point, and indeed, this is an area where we have shortcomings. We have added information about disease specificity in agricultural production practices in the "Limitations and Future Work" section. The specific details are as follows: On the other hand, due to the highly complex nature of field environments, diseased plants may be subjected to various types of stressors, such as drought, flooding, nutrient deficiencies, and more. Additionally, for the plants themselves, they may not only be affected by a single disease during their growth process; the occurrence of multiple diseases concurrently is something we have not thoroughly considered.

  1. Reviewer #3:In “Limitation and future work” authors wrote: “We aim to acquire data on peanut southern blight from multiple provinces and varieties to enhance the model”. In my opinion the future work should be extended to evaluation of different stress types (different diseases symptoms, water stress, nutrient deficiency symptoms, …) not only limited to one disease.

Response:Thanks to the expert's comments. Thank you very much for your valuable feedback. In response to this issue, we have supplemented the following information in the "Limitations and Future Work" section: At the same time, we need to further refine our experimental design, make fuller use of experimental resources, systematically control variables, and expand future work to assess the effects of various stressors.

Round 2

Reviewer 3 Report

Comments and Suggestions for Authors

The changes in the manuscript and explanations of the authors are sufficient for publication.